

# Finding Gaia: Exploring Climate Change Through Gamification

Maria Vittoria Gargiulo[1*], Raffaella Russo[1], Paolo Capuano[1]

[1] Dipartimento di Fisica "E.R. Caianiello", Università degli Studi di Salerno, Fisciano (SA), Italia

*Correspondence to*: Maria Vittoria Gargiulo (mgargiulo@unisa.it)

**Abstract.** Effective science communication is a vital tool in bridging the divide between scientific progress and the well-being of society, ensuring that the fruits of research are not only accessible but also comprehensible to the broader public. By tailoring communication strategies to different audiences, we can foster greater engagement and facilitate a deeper understanding of complex topics. In particular, involving young people in science communication is crucial, as it not only promotes innovation but also empowers them to tackle pressing global challenges.

Gamification has emerged as an innovative approach in this context, incorporating game-like elements to captivate and educate audiences. Platforms such as Kahoot! and Quizizz are widely recognised for enhancing learning motivation, while crowd-based games like Foldit are revolutionising scientific research by harnessing the power of collective intelligence. When applied to climate change education, gamification proves particularly effective, creating a platform for both deepening understanding and driving proactive behavioural change.

In response to the challenges posed by the COVID-19 pandemic, virtual platforms became indispensable in maintaining science communication efforts. A prime example of this adaptation is the creation of "Finding Gaia" – an immersive educational experience focused on climate change. This initiative, informed by established evaluation protocols, was designed to assess its impact on participants. Statistical analysis revealed significant knowledge gains, underscoring the effectiveness of the gamified approach in achieving its educational objectives.

## 1    Introduction

Science communication serves as a cornerstone in translating research into meaningful societal benefits. Its importance lies in narrowing the divide between scientific innovation and its practical applications in everyday life (Bucchi 2008). Effective science communication ensures that research findings extend beyond academic circles, becoming accessible and

comprehensible to broader audiences, thereby enabling informed decision-making and advancing societal progress (Besley & Tanner 2011, Nisbet & Scheufele 2009, Scheufele & Lewenstein 2005).

To address the diverse nature of society, developing multi-layered and inclusive communication strategies is essential (Brossard & Nisbet 2007, Stilgoe et al. 2014). Different societal groups exhibit varying levels of scientific literacy, interests, and needs (European Commission 2007). By tailoring communication approaches to meet these diverse requirements, we can



enhance engagement, understanding, and relevance of scientific information (Foundation for Science and Technology 2019, European Science Communication Network 2020). For instance, employing simplified language and visuals may resonate more effectively with lay audiences, while more technical discussions could cater to policymakers and domain experts (Foundation for Science and Technology 2019).

Young people hold a pivotal role in shaping future solutions to global challenges (Mitchell et al. 2008, OCED 2020). Equipping

youth with scientific knowledge and critical thinking skills empowers them to actively address pressing societal issues (Jun et al. 2021, UNESCO 2022). Engaging young audiences in science communication activities fosters a generation of scientifically literate individuals capable of driving innovation and societal progress (Fernandez & Shaw 2013). Additionally, their fresh perspectives and creative problem-solving capabilities can inspire novel approaches to complex issues.

Gamification represents an innovative approach to science communication, leveraging game design principles to captivate and

educate audiences (Hamari et al. 2014, Poushter et al. 2018). Its immersive and interactive features capture attention, facilitate learning, and encourage participation (Deterding et al. 2011, Baranowski et al. 2008). By transforming traditional methods of conveying complex information, gamification not only enhances user engagement but also helps distill intricate scientific ideas into accessible, dynamic experiences. This adaptability makes it particularly valuable in contexts where conventional communication techniques may fall short, effectively bridging the gap between specialized knowledge and general

understanding.

This approach has proven effective across various fields, including education, healthcare, and environmental conservation (Arnold et al. 2012). In education, platforms like Kahoot! (Arnesen 2019, Hew & Cheung 2018) and Quizizz (Sinha et al. 2018) gamify learning, transforming quizzes into competitive games that boost student motivation and knowledge retention. Similarly, Foldit (Cooper et al. 2010, Khatib et al. 2011), a game where participants solve protein-folding puzzles, demonstrates

how gamification can crowdsource scientific research and lead to groundbreaking discoveries. Applications like Duolingo apply gamified elements to language learning, using rewards such as points and levels to motivate users (Anderson 2016). Fitness apps (Cavallo et al. 2012, Patel et al. 2015) incorporate mechanisms like level progression and user challenges to encourage healthy behaviours.

Moreover, the integration of gamified elements promotes not only individual learning but also collaborative problem-solving

and continuous self-improvement. By offering immediate feedback, clear goals, and measurable progress, gamification fosters an environment where users feel empowered to explore, experiment, and master new skills. This multifaceted approach underscores its potential to revolutionize how we communicate science, making it an essential tool for engaging diverse audiences and driving meaningful outcomes (Hamari et al. 2014).

Climate change education presents a compelling avenue for gamification, given its interdisciplinary nature and critical

importance (Lillig & Guha 2017, Woo & Lee 2015). This method simplifies complex concepts, making them accessible through interactive games that foster an understanding of climate systems (Fernández Galeote & Hamari 2021, Gerber et al. 2021, Douglas & Brauer 2021). These experiences not only educate participants but also empower them to take meaningful action within their communities (DeWaters & Powers 2011, Kirchner et al. 2015).



In this context, during the academic year 2020/21, the constraints of the COVID-19 pandemic necessitated the transition of
our outreach and engagement initiatives from in-person laboratories to virtual platforms. Adapting to the virtual tools approved
by schools, we aimed to retain an informal yet engaging learning experience. Our primary goal was to position students as
active participants, fostering scientific understanding alongside leadership and problem-solving skills.

To achieve these aims, we employed gamification, recognising its potential to deliver effective and enjoyable learning
experiences across varied genres, technologies, and demographics. Accordingly, we developed a climate change-focused
educational activity, "Finding Gaia," with the aim of establishing a best-practice model for science communication.

Drawing from the work of Jensen & Gerber (2020) and Veldekamp et al. (2021), we designed an evaluation framework to
measure the impact of our serious games in teaching and communicating the concept of risk. This paper presents the results
obtained from this evaluation process.

## 2    Methods

### 2.1    Gamification

Gamification was first formally introduced by Nick Pelling, a British entrepreneur, in 2012 (Pelling 2012). Although the term
was not widely recognised at the time, it originated from the concept of using game elements and principles in non-gaming
contexts to drive motivation, engagement, and learning. The foundational work in defining and exploring gamification as a
practice was conducted by Deterding et al. (2011), who established its characteristics and potential applications. Since then,
gamification has gained prominence, particularly from the 2010s onwards, driven by the rise of digital technologies and an
increasing emphasis on user engagement and experience (Huotari & Hamari 2012). The theoretical underpinning of
gamification lies in psychological and motivational principles, such as instant gratification, social competition, and the drive
to achieve personal goals. These principles are employed to influence user behaviour and encourage specific actions (Deterding
et al. 2011, Hamari et al. 2014, Zichermann & Cunningham 2011). It is critical to note that gamification does not transform
activities into full-fledged games but integrates selected gaming elements to enrich the user experience (Gerald 2018). The
primary objectives of gamification include enhancing motivation, adherence, and productivity (Deterding et al. 2011, Hamari
et al. 2014). Key mechanisms of gamification include awarding points, badges, or virtual rewards for completing tasks,
fostering challenges and competitions among users, and implementing systems for advancement and progression (Deterding
et al. 2011, Hamari et al. 2014, Nicholson 2012).

### 2.1.1    Gamification in Climate Change Education

Gamification provides a promising avenue for climate change education by captivating and motivating learners through
interactive and immersive experiences (Lillig & Guha 2017). By transforming intricate scientific concepts and environmental
challenges into engaging games and simulations, gamification makes learning about climate change more accessible,
enjoyable, and meaningful for diverse audiences (Woo & Lee 2015). This approach allows players to explore the consequences





of their environmental decisions interactively, fostering a deeper understanding of the interconnectedness of climate systems and human actions (Fernández Galeote & Hamari 2021). Games simulating real-world scenarios enable players to test strategies for mitigating and adapting to climate change, empowering them to make informed choices and take meaningful actions within their communities (Gerber et al. 2021, Douglas & Brauer 2021). Gamification encourages learning and behaviour change through mechanisms like rewards, competition, and progression systems (DeWaters & Powers 2011, Kirchner et al. 2015). By earning points, unlocking achievements, and competing with peers, players are incentivised to understand climate-related concepts, adopt sustainable behaviours, and champion environmental stewardship. In addition to these mechanisms, several gamified interventions specifically targeting climate change education have been developed. For instance, Parker et al. (2016) designed a game to engage stakeholders in extreme event attribution science, offering a practical framework to explore the uncertainties and risks associated with climate extremes. Similarly, Mendler de Suarez et al. (2012) introduced Games for a New Climate: Experiencing the Complexity of Future Risks, which immerses players in the multifaceted challenges of future climate scenarios, thereby deepening their appreciation of climate complexity. Furthermore, the World Climate Simulation provides an interactive, role-based simulation that places participants in the midst of international climate negotiations, illustrating the intricate dynamics of global decision-making in response to climate change (World Climate Simulation).

## 2.2    Developing Finding Gaia

Building on this methodological foundation, we designed *Finding Gaia*, a virtual treasure hunt employing gamification techniques. The initiative aims to engage participants while encouraging them to explore fundamental concepts of geophysics and climatology through play. The activity focuses on climate change, including mitigation and adaptation strategies, with the dual objective of raising awareness about the energy transition and inspiring the younger generation to pursue science, particularly geophysics.

Designed primarily for secondary school students and science enthusiasts, *Finding Gaia* employs a virtual and inclusive approach, ensuring accessibility for individuals with reduced mobility. The activity integrates interactive elements, including quizzes, puzzles, and tasks of varying difficulty, to engage both novice and advanced participants. This design facilitates not only top-down learning but also peer-to-peer knowledge sharing. The team-based format promotes soft skills such as leadership and collaboration, while also fostering individual problem-solving abilities.

Participants, whether solo or in teams, progress through the treasure hunt by unlocking content at each stage. Each step introduces new concepts related to climate, enabling both game advancement and learning. The activity, which spans approximately 120 minutes, is guided by experts who provide critical insights throughout the experience. This expert guidance ensures participants achieve a nuanced understanding of the topics covered.





### 2.2.1 Key Gamification Features of *Finding Gaia*

*Finding Gaia* incorporates several hallmark features of gamification to enhance engagement and learning (Deterding et al. 2011, Hamari et al. 2014):

1. **Points and Rewards:** Participants earn points or rewards for completing tasks, solving puzzles, or uncovering virtual treasures. These rewards contribute to leaderboard rankings, fostering competition.
2. **Clues and Challenges:** Participants receive clues and challenges in various formats—riddles, puzzles, and more—that guide them through the content, adding excitement and intrigue.
3. **Leaderboards:** Rankings displayed on leaderboards motivate participants to compete and track their performance.
4. **Social Interaction:** Live chats and messaging allow participants to collaborate, share tips, and celebrate milestones, enhancing the social dimension of the activity.
5. **Progress Tracking:** Participants can monitor their progress, helping maintain focus and motivation throughout the treasure hunt.

## 2.3 Evaluation

The evaluation of science communication activities is essential for understanding their effectiveness in engaging stakeholders and fostering scientific literacy (Jensen & Gerber, 2020). Drawing inspiration from Veldkamp et al. (2021), who examined the educational potential of escape rooms in science education, we employed a combination of surveys and qualitative classroom observations to evaluate students' perceptions of the activity's educational value. This mixed-methods approach also informed the interpretation of quantitative survey data.

Prior to and following the activities, students completed a structured survey (Appendix A) designed to assess their baseline knowledge, interest levels, and expectations, as well as the overall impact of the event. The survey incorporated multiple-choice questions, Likert scales (Likert, 1932), and open-ended questions to capture both quantitative and qualitative feedback. Likert scales, widely used in psychometric research, enable the measurement of attitudes towards specific objects, events, or concepts, using a numerical scale—commonly ranging from 1 to 5. The inclusion of multiple-choice and open-ended questions further enriched the qualitative insights gathered from participants.

To ensure ethical data collection practices, all responses were anonymised, and no personal sensitive information, such as age, gender, or religion, was collected except in aggregated form. Participation in the survey was entirely voluntary, with no obligation to answer any question that could potentially cause discomfort.

To evaluate the assimilation of concepts presented during the activity, a t-Student test (Student, 1908) was performed to analyse statistical correlations between pre- and post-activity responses. With a 99% confidence level and 206 degrees of freedom, the analysis confirmed statistically significant differences in mean values before and after the session, demonstrating the educational impact of the experience. Additionally, Cronbach's alpha (Cronbach, 1951) was employed to assess the internal



consistency of the survey data, measuring the degree of association among variables and comparing Likert scale mean values across the pre- and post-surveys.

While the described evaluation method offers meaningful insights, we recognise several limitations inherent to the process, primarily due to the constraints of the COVID-19 pandemic. To mitigate the potential burden of prolonged screen time on students, the survey was deliberately simplified. Moreover, the relative infancy of this field posed challenges in benchmarking our methodology against existing literature.

A distinctive feature of this experience lies in its non-replicable nature, a characteristic that underscores the uniqueness of this
work. Despite the potential for survey improvements, this aspect contributes significantly to the study's originality, offering a novel perspective in the exploration of innovative approaches to science communication.

## 3  Results

"Finding Gaia" was initially implemented in collaboration with the University of Naples Federico II, the University of Sannio,
and the INGV during the 2021 edition of the science fair *Futuro Remoto*. Subsequently, it was replicated during the 2021/22 iteration of the *Piano Lauree Scientifiche* (PLS – Scientific Degree Plan) at the Department of Physics "E.R. Caianiello" of the University of Salerno, as well as within the framework of the project *IDEE – Institution of a Deal for Environmental Education*. This initiative, led by the Department of Chemistry and Biology "A. Zambelli" of the University of Salerno, aimed to establish a partnership between schools and universities to enhance, innovate, and disseminate a scientific culture oriented
towards environmental awareness."

The data analysed in this study were gathered during the *IDEE* project, which involved the participation of 206 students in the *Finding Gaia* activity. To ensure the internal consistency of the collected data and to evaluate the degree of association among variables as a cohesive metric, Cronbach's alpha was calculated for both the pre- and post-evaluation datasets. As shown in Table 1, the results (Cronbach's alpha > 0.70) indicate a high degree of reliability and internal consistency among the
indicators (Cronbach, 1951).

These metrics provided a robust basis for comparing mean values from the Likert scales in the pre- and post-evaluation surveys. The subsequent sections will present the results derived from this comparison, highlighting the impact of the activity on participants' knowledge and perceptions.

### 3.1  Quantitative Results

Figure 1 presents the key qualitative characteristics of the participating students. The pie charts indicate that 64% of participants were aged 16–17, 30% were 18 years old, and only 6% were 15 years old, with an approximately equal distribution of genders. Notably, the majority of students reported no prior experience with recreational or educational escape rooms; in fact, only 9% had participated in a treasure hunt before.



Figures 2 through 6 illustrate the distribution of responses to control questions administered before and after the protocol. The

Likert scale used for these responses, ranging from 1 to 5, is detailed in Appendix A. The figures demonstrate a shift towards higher ratings across response distributions, suggesting a favourable impact of the protocol. To substantiate this observation and determine whether students assimilated the concepts presented, statistical correlation analyses of pre- and post-experience responses were conducted using a t-Student test to assess the significance of distribution differences. These results are summarised in Table 2.

For each survey question, the results indicate a rejection of the null hypothesis with a 99% probability ($p < 0.01$) for Questions A to D, and a 95% probability ($p < 0.05$) for Question E, with 206 degrees of freedom. This confirms a statistically significant difference between the pre- and post-protocol distributions. The implications of these differences are examined for each question as follows:

Question A: How would you rate your knowledge on climate change?

Before the protocol, the average rating for knowledge on climate change was 3.2, indicating a moderate level of understanding. After the protocol, this rating increased to 3.9, suggesting that participants perceived the activity as both educational and effective. - Figure 3

Question B: How would you rate your awareness of the risk posed by climate change in your region?

Prior to the protocol, the average rating for awareness of regional climate change risks was 3.4, slightly above neutral.

Following the protocol, this rating rose to 4.0, reflecting an increased awareness. Although this finding underscores the perceived efficacy of the protocol in highlighting climate change risks, it is important to recognise that the measure reflects self-perception rather than objective knowledge. – Figure 4

Question C: How interested are you in geophysics?

Initially, the average interest in geophysics was rated at 3.0, indicating a neutral level of interest. Post-protocol, this increased

to 3.6, suggesting that the activity fostered greater interest in the subject, potentially due to the engaging nature of the protocol. – Figure 5

Question D: How interested are you in environmental science?

Before the protocol, interest in environmental science had an average rating of 3.4, ranging from neutral to moderately high. After the protocol, this increased to 3.8, signifying an enhanced interest, thereby highlighting the positive impact of the protocol

on student engagement with environmental science topics. – Figure 6

Question E: How much and how do you think the virtual characteristic of the treasure hunt will affect the experience?

Regarding the influence of the virtual format on the overall experience, the pre-protocol average rating was 3.6, reflecting neutral to slightly positive expectations. Following the protocol, this rating increased to 3.8, indicating a more favourable perception. This suggests that integrating gamification elements into the virtual treasure hunt was regarded as a beneficial and

appealing feature by the participants. – Figure 2



### 3.1.1 Qualitative Results

The post-protocol questionnaire included supplementary questions designed to evaluate students' perceptions of their learning outcomes during the activity. Specifically, participants were asked to reflect on the educational objectives they believed were achieved through their engagement with the escape room. The results, presented in Figure 8, indicate that students perceived the experience as highly positive. They reported not only acquiring new knowledge and skills but also enhancing and applying previously learned concepts. Moreover, the activity was recognised as beneficial for fostering team-building skills and motivating further academic interest in geophysics and environmental sciences.

To evaluate the overall appeal of the protocol, standard marketing questions were employed to measure retention and advocacy, both in relation to the protocol itself and its perception as "the product." Figure 7 and Table 3 summarise the outcomes pertaining to likeability, retention, and advocacy.

## 4 Conclusion & discussion

Science communication holds a pivotal role in shaping a culture of risk awareness, especially in an era marked by growing mistrust towards scientific institutions (Algan et al., 2021). By engaging young people—our future leaders—in these efforts, we not only enhance scientific literacy but also empower them to act as knowledge brokers within their families and communities (Mitchell et al., 2008; OECD, 2020). Equipping the younger generation with robust scientific knowledge and critical thinking abilities positions them to address pressing societal challenges (Jun et al., 2021; UNESCO, 2022). More than that, it fosters a cohort poised to champion innovation and drive transformative change (Fernandez & Shaw, 2013). The creativity and novel perspectives of youth, unencumbered by traditional constraints, can spark new solutions to the complex problems of our time.

One such emerging strategy in science communication is gamification. By infusing elements of game design into educational efforts, it not only captures attention but also nurtures engagement and deeper learning (Hamari et al., 2014; Poushter et al., 2018). Its interactive and immersive qualities encourage active participation, making the process of learning both enjoyable and memorable (Deterding et al., 2011; Baranowski et al., 2008). Already, gamification has proven its worth in diverse domains, from education to healthcare, and even in environmental conservation (Arnold et al., 2012).

With this in mind, we developed a novel educational protocol targeted at high school students (ages 15–18+), which transformed a lesson on climate change and its mitigation and adaptation strategies into an engaging virtual treasure hunt. This gamified activity was designed not only to impart knowledge on seismic risks and climate change but also to spark greater interest in the fields of environmental science and geophysics.

Our findings have been promising. Students reported that the activity was both effective and enjoyable, showing a notable increase in their enthusiasm for science and geophysics. Many qualitative responses highlighted a newfound appreciation for the relevance of science in everyday life, pointing to an increase in their Science Capital. For example, one student commented, "*I never realized that science is actually everywhere and affects what we do every day.*" Another one, "*This activity made*



*science feel more real to me."* These typical responses help illustrate the depth of the students' engagement and the significant shift in their perception of science as both accessible and important for understanding real-world problems. These qualitative

insights further complement the quantitative trends shown in Figures 7 and 8, reinforcing the overall impact of the activity.

We argue that single-session activities, such as the one tested in this study, can serve as powerful tools to raise awareness, offering experiences that are not only impactful but also memorable. When these are complemented by sustained engagement over a longer period, they can further enhance understanding and retention of complex topics. Gamification, by making the learning experience interactive and engaging, holds considerable potential in addressing difficult subjects like climate change,

fostering both comprehension and active participation.

Nevertheless, it is essential to acknowledge the limitations of our approach and the ongoing need for its evaluation and refinement to suit various educational contexts and diverse learning styles. Future research should explore the long-term effects of gamification on knowledge retention and behavioural change, as well as identify the optimal design elements and implementation strategies for integrating such methods into formal education settings (Lillig & Guha, 2017; Woo & Lee,

2015). Additionally, longitudinal studies and innovations that enhance scalability and accessibility should be prioritised.

The success of the protocol can be partly attributed to its suitability for the target audience, as the challenges and content involved a level of logical and mathematical reasoning typically developed at the high school stage. However, despite the valuable insights gathered through our evaluation, several limitations must be noted, particularly those arising from the constraints imposed by the COVID-19 pandemic. In order to mitigate the potential burden of prolonged screen time for

students, the survey was intentionally simplified. Furthermore, due to the emerging nature of this field, benchmarking our methodology against established literature presented some challenges, such as the limited availability of standardized metrics, inconsistencies in key definitions, and variations in research approaches.

An intriguing feature of this project is its non-replicable nature, which lends it a unique character. While improvements in survey design are certainly warranted, the singular nature of this study contributes significantly to its originality and offers a

fresh perspective on innovative methods for science communication.

Looking ahead, we intend to examine how factors such as gender influence the outcomes, involve the families of participating students, and account for socio-economic and psychological factors that may shape the results. In conclusion, while the protocol has proven successful, future iterations will delve deeper into dimensions such as gender dynamics, family involvement, socio-economic context, and psychological considerations to further enhance its inclusivity and overall

effectiveness in science communication.





## 5   Tables

**Table 1 - Cronbach Alpha results**

| Cronbach Alpha | |
|---|---|
| PRE | **0,8** |
| POST | **0,8** |

**Table 2 - t Student test result comparing pre and post protocol**

| | PRE | | POST | | p value |
|---|---|---|---|---|---|
| | **μ** | **σ** | **μ** | **σ** | |
| Question A | 3,2 | 0,8 | 3,9 | 0,8 | **4E-20** |
| Question B | 3,4 | 0,9 | 4,0 | 0,9 | **3E-11** |
| Question C | 3,0 | 1,0 | 3,6 | 1,0 | **3E-11** |
| Question D | 3,4 | 1,1 | 3,8 | 1,0 | **4E-07** |
| Question E | 3,6 | 0,8 | 3,8 | 1,0 | **3E-02** |

**Table 3 – Likeability, Retention & Advocacy**

| | **μ** | **σ** |
|---|---|---|
| Likeability | 4,5 | 0,7 |
| Retention | 4,3 | 0,9 |
| Advocacy | 4,2 | 1,0 |

## 6   Figures






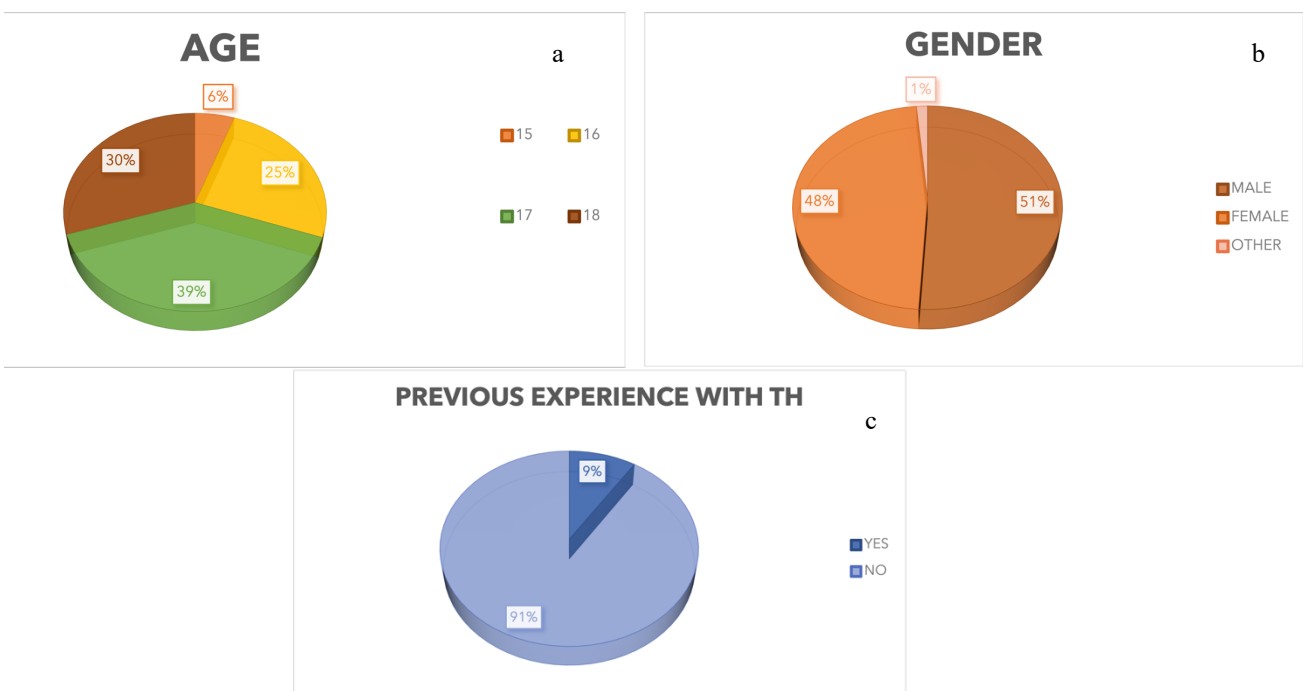

**Figure 1 - Demographic characteristics and prior experience of the participating students. The pie charts indicate that 64% of the students were aged 16–17, 30% were 18 years old, and 6% were 15 years old, with an approximately equal gender distribution. In addition, only 9% of the participants reported previous experience with recreational or educational escape rooms.**




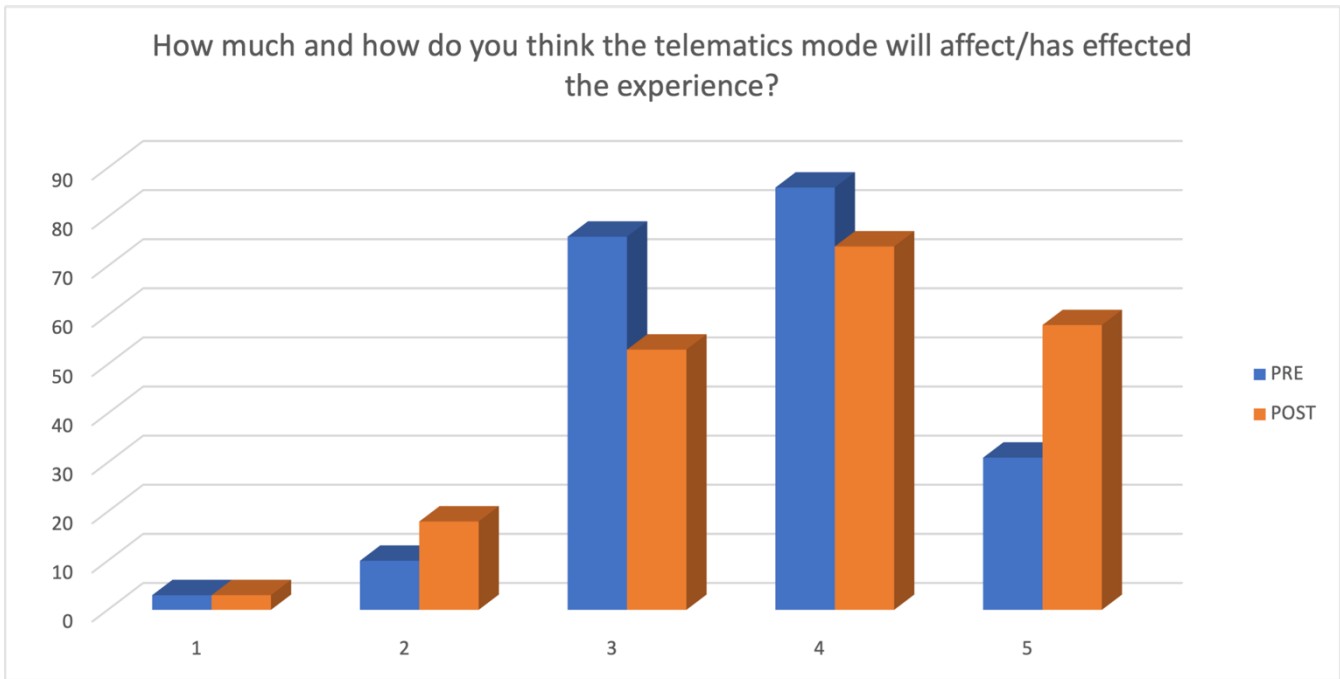

**Figure 2 - Distribution of responses to Question E: "How much and how do you think the virtual characteristic of the treasure hunt will affect the experience?" The figure compares pre-protocol (mean = 3.6) and post-protocol (mean = 3.8) ratings on a 5-point Likert scale.**





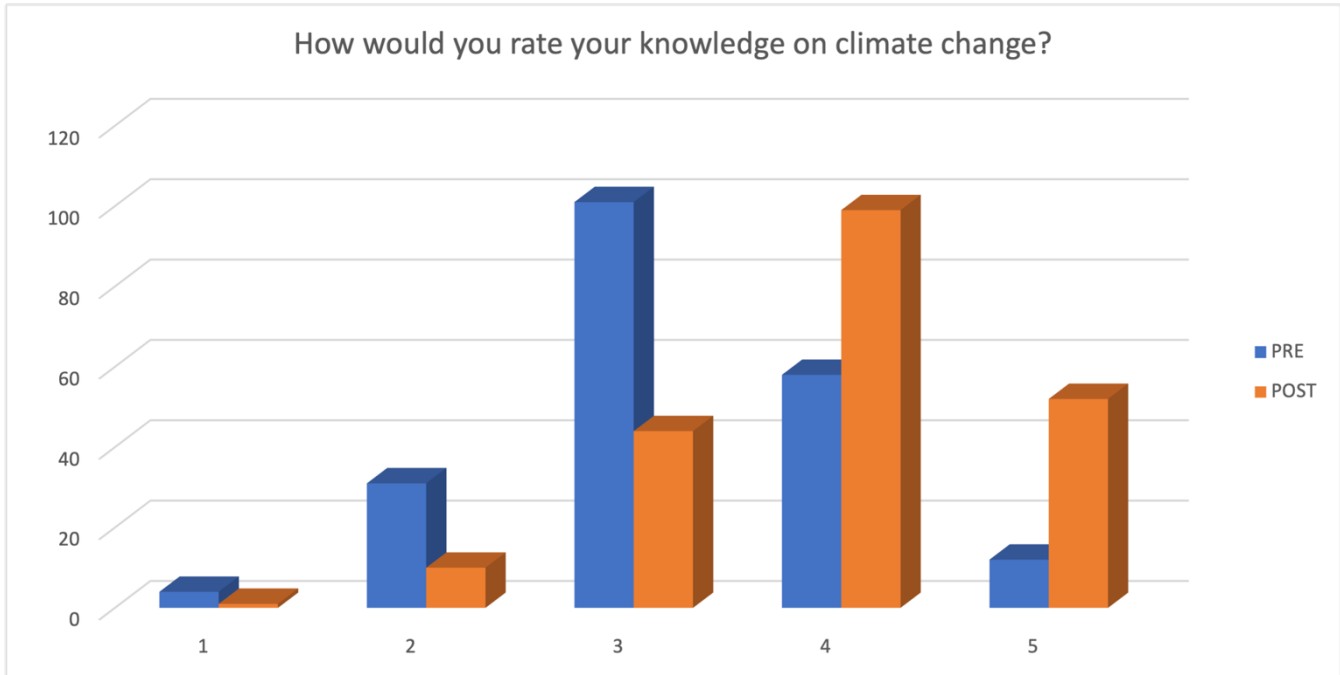

**Figure 3 - Distribution of responses to Question A: "How would you rate your knowledge on climate change?" The histogram shows an increase in the average rating from 3.2 (pre-protocol) to 3.9 (post-protocol) on a 5-point Likert scale, indicating improved perceived knowledge.**



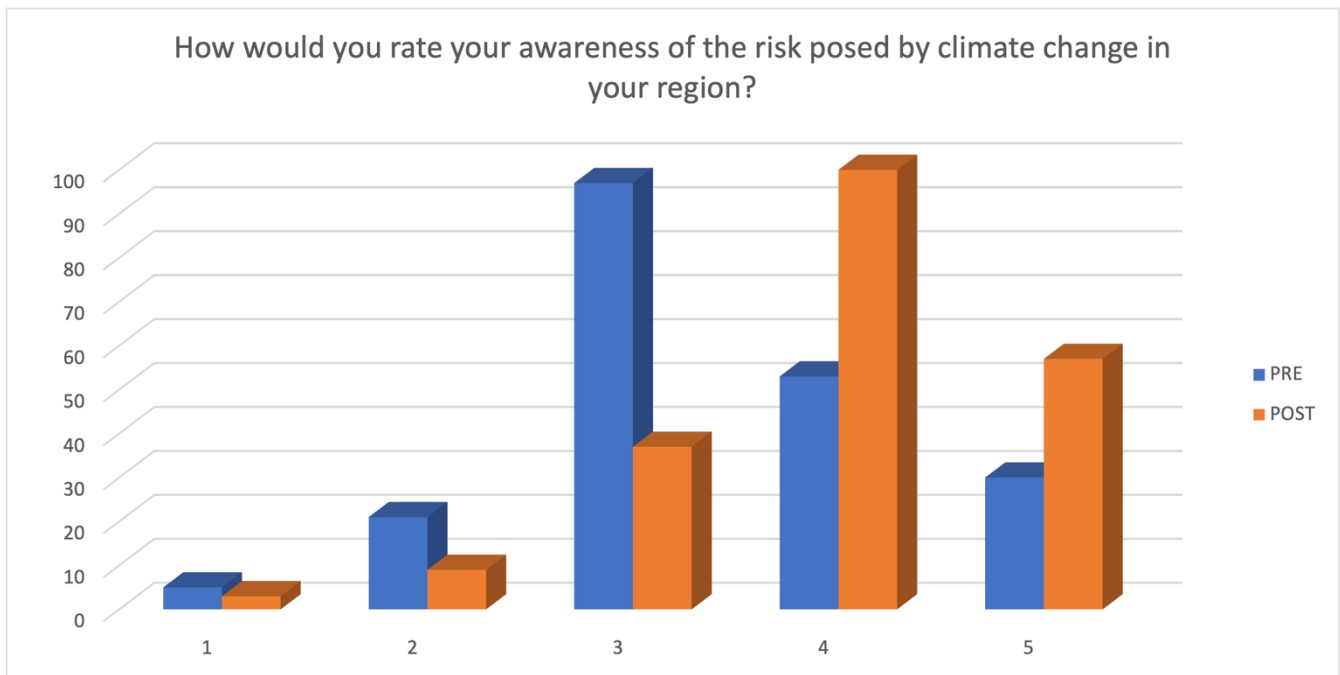

**Figure 4 - Distribution of responses to Question B: "How would you rate your awareness of the risk posed by climate change in your region?" The data illustrate an increase in the average rating from 3.4 pre-protocol to 4.0 post-protocol, as measured on a 5-point Likert scale.**


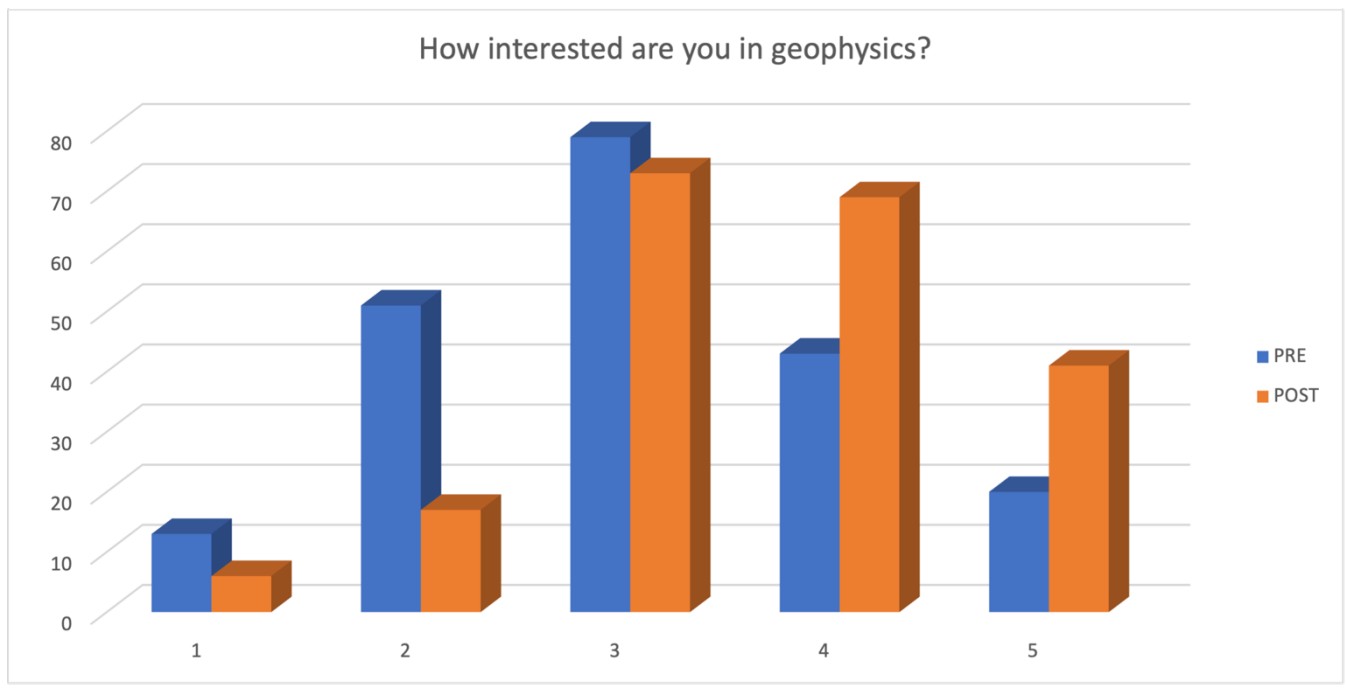

**Figure 5 - Distribution of responses to Question C: "How interested are you in geophysics?" The histogram reflects an increase in average interest from 3.0 (pre-protocol) to 3.6 (post-protocol) on a 5-point Likert scale.**



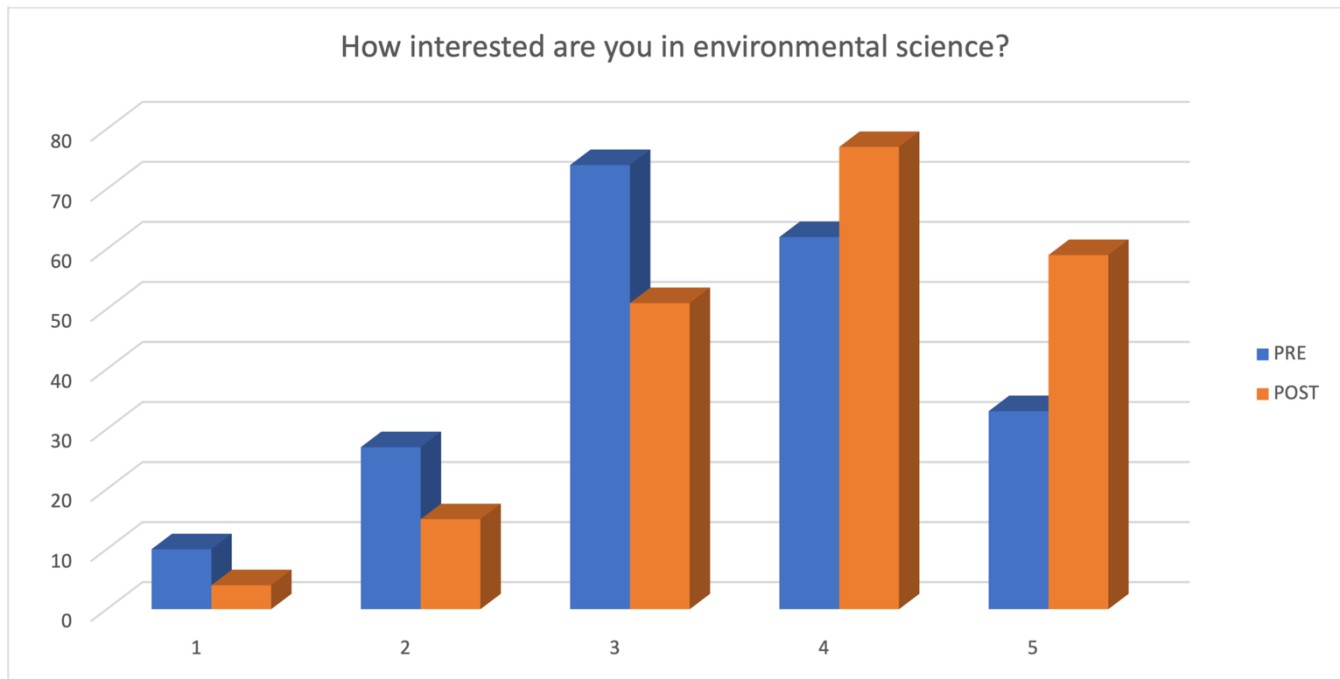

**Figure 6 - Distribution of responses to Question D: "How interested are you in environmental science?" The figure demonstrates a shift in the average rating from 3.4 (pre-protocol) to 3.8 (post-protocol) on a 5-point Likert scale, indicating enhanced interest following the protocol.**

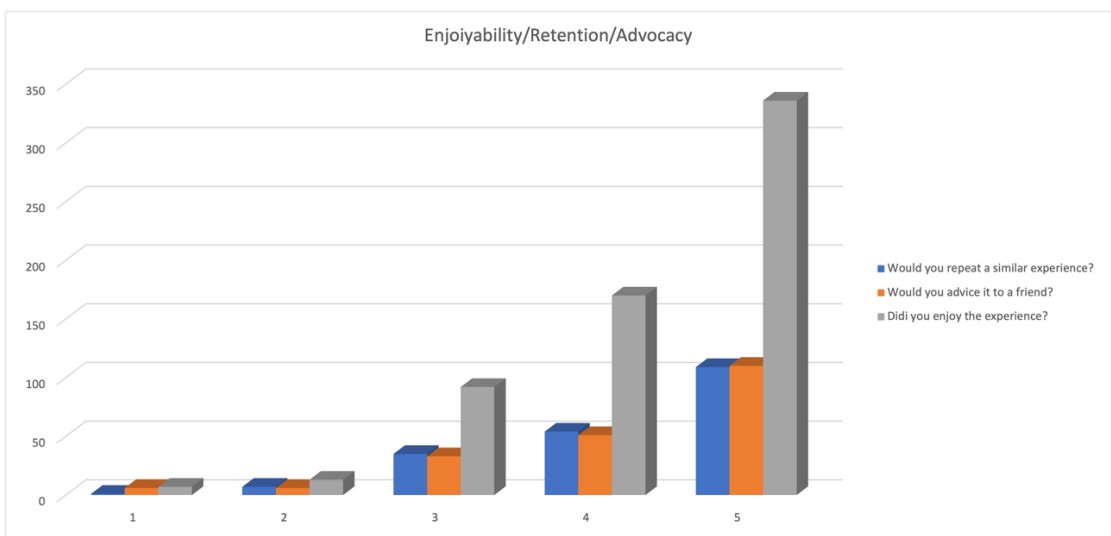

**Figure 7 - Distribution of outcomes pertaining to likeability, retention, and advocacy. This histogram illustrates participants' post-protocol ratings on a 5-point Likert scale for three key dimensions: the overall likeability of the activity, the retention of the educational content, and the likelihood of advocating the activity to others. The figure shows that the majority of respondents provided high ratings across these dimensions, suggesting that the activity was not only enjoyable but also memorable and recommendable**




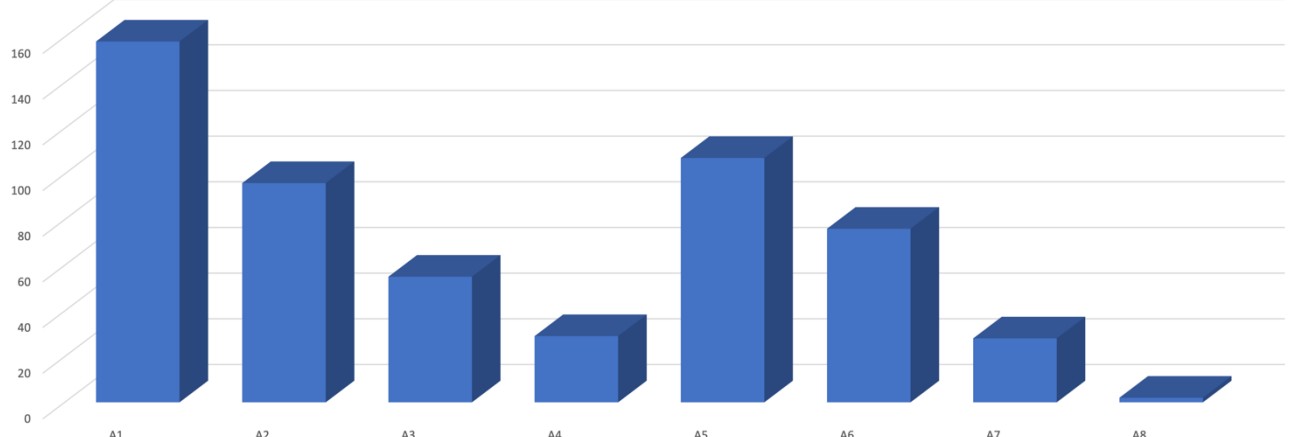

**Figure 8 - The histogram displays the frequency distribution of responses to the multiple-choice question: "What do you think were the educational goals achieved with the treasure hunt?" The x-axis represents the eight distinct answer options: A1: Acquisition of new content knowledge and skills; A2: Developing content knowledge and skills; A3: Testing content knowledge and skills; A4: Formative assessment; A5: Improving teamwork; A6: Improving motivation for geophysics; A7: Getting to know each other; A8: Other objective(s). The y-axis indicates the number of respondents who selected each option. Each bar's height corresponds to the frequency with which participants identified that specific educational goal.**

## 7   Ethical statement

The research conducted for the paper adheres to the highest ethical standards. The study design and data collection procedures were meticulously planned and executed to ensure the privacy, anonymity, and voluntary participation of all subjects involved. All data collected during the evaluation were anonymized to protect the identity of the participants. No identifying information was retained or associated with the responses. The collected data did not include any sensitive personal information such as age, gender, religion, or other identifiers, except when such information was aggregated and used for statistical analysis without linking to individual participants. Participation in the study was entirely voluntary. Participants were informed that they could choose not to answer any question that they felt uncomfortable with or that could potentially offend them in any way. Clear instructions and assurances were provided to participants, emphasizing that their choice to participate or not, as well as their responses, would not affect them adversely in any manner. Prior to data collection, participants were provided with detailed information about the purpose of the study, the nature of their involvement, and the measures taken to ensure their anonymity and data protection. By adhering to these ethical principles, we ensured the integrity of our research process and the protection

of our participants' rights and well-being. We are committed to maintaining these high standards in all our research activities and welcome any questions or concerns regarding our ethical practices.

## 8  Conflict of Interest

The authors declare that the research was conducted in the absence of any commercial or financial relationships that could be construed as a potential conflict of interest.

## 9  Author Contributions

M.V.G. has developed the Treasure Hunt and the evaluation protocol and has carried on the data analysis. M.V.G., R.R., P.C. contributed in the activity and reviewed the work. P.C. is responsible for funding.

## 10  Funding

This work has been partially supported by the project CORE – sCience and human factor for Resilient SociEty founded by the
European Union's Horizon 2020 research and innovation program under grant agreement No 101021746.

## 11  Acknowledgments

Finding Gaia was firstly carried out in collaboration with the University of Napoli Federico II, the University of Sannio and the INGV during the 2021 Edition of the Science Fair "Futuro Remoto". It was then repeated during the 2021/22 edition of the *Piano Lauree Scientifiche* – PLS (Scientific Degree Plan) at the department of Physics "E.R. Caianiello"of the University
of Salerno and for the project "IDEE - Institution of a Deal for Environmental Education: Istituzione di un accordo tra scuola e università per il potenziamento, l'innovazione e la divulgazione di una cultura scientifica orientata all'ambiente" of the department of Chemistry and Biology  "A. Zambelli" of the University of Salerno.
We acknowledge all the people and researchers involved in these three endeavours. A particular mention goes to Dr. Ferdinando Napolitano and Dr. Ortensia Amoroso, members of the Geophysics and Seismology Laboratory of the Department
of Physics "E.R. Caianiello" of the University of Salerno.

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
