# Peer review of "Finding Gaia: Exploring Climate Change Through Gamification"

_EGUsphere, 2025_

## Referee Comment (RC1)

**Review of **egusphere-2025-577** – Finding Gaia (FG)**

By David Crookall, supported by Pimnutcha Promduangsri

This review is in several parts:

**Overview - courage**

We would like to warn you that this review contains many criticisms. We urge you to take them as ways to help you improve. Please do not be discouraged by the number and type of comments; they are there to help. We believe that, with much work and many tweaks, you will be able to turn your ms into a publishable article. It does mean a lot of work, but it is really worthwhile work. You should not be in a rush, but focus on quality. Then you will have an article that is a good contribution to the literature and is likely to be cited by other scholars. Have courage and persistence.

**Comments on game concepts and claims**

| | |
|---|---|
| Terms | In the manuscript (ms), you use a wide range of terms related to simulation/gaming or gaming (for short), (a term that is often/sometimes used to denote the whole area of games, simulations, role-play, THs, exercises, ludic activity, etc.). The messy use of terminology in the literature should not be allowed to creep into your scientific article. In particular, the unhappy term 'serious game' should be avoided, especially as, after a faltering and potted history, it seems to be going out of fashion; in any case, its meaning has changed radically since its first appeared – I have provided an argument here https://www.researchgate.net/publication/374344073_Debriefing_A_Practical_Guide, p.124. |
| | We suggest that, fairly early on, maybe in the introduction, you provide a glossary of the terms that you will use in the ms. This is not an easy task, given the indiscriminate use of a whole host of terms in the media, including academic journals. You will find references to fairly reliable sources at the end of this review. |
| Treasure hunt = game | It is only after reading someway into your ms that we discover that FG is a treasure hunt (TH) and not a gamification method – a TH during which gamification techniques are deployed. (Correct me if I am wrong). In fact, a TH is (to all intents and purposes) a game. You yourself say so in line 122 "progress through the **TH** by unlocking content at each stage. Each step introduces new concepts related to climate, enabling both **game** advancement and learning". Here are some arguments that you can use to 'claim' that a TH is a game: |
| | ○ Goal: Just like most games, a TH has a clear objective: to find the hidden treasure. |

- Rules: While they might not be as formal as a board game's rulebook, THs typically have rules. These could involve following clues in a specific order, staying within designated boundaries, or solving puzzles in a particular way.

- Competition: While not always explicitly competitive, we often have an inherent element of competition in both games and THs. Individuals or teams race against each other to be the first to find the prize. (Note that some games are designed specifically to be cooperative.)

- Engagement: At its core, a TH is usually designed to be enjoyable and engaging – see Whiton's excellent work on engagement. The acts of deciphering clues, exploring different locations and the anticipation of discovery all contribute to an engaging experience.

- End criteria: Games typically have a clear ending, and a TH is the same. The game concludes when the treasure is found.

- Debriefing: Both games and THs should (must) be thoroughly debriefed for substantial learning to take place – see the debriefing guide ref'd above.

| | |
|---|---|
| Concepts & games | "By transforming intricate scientific concepts and environmental challenges into engaging games and simulations, gamification makes learning about climate change more accessible, …" (l.92). I do not see how gamification can transform concepts into simulation/games. Concepts are transformed through the game design process. |
| 91 | "Gamification provides a promising avenue for climate change education". It is not like climate change (CC) edu is looking around for promising avenues; gamification is just one method that seems to have some potential to increase CC literacy. However, I would suggest that it is not gamification per se, but the encapsulating activity (eg, your FG TH), in which certain elements are gamified, that provide the literacy. |
| Refs & claims | You mention a number of works on gamification. However, you should mention, and maybe quote from, others, including those who detract from the idea that gamification is the best thing since sliced bread. See the refs below. It is also prudent to be wary of wild claims that some gamers make about their games, work and gamification hooks.

Most gamers, including ourselves (you and us), are keen on simulation/gaming and are convinced that they can work – to a large degree and in certain circumstances – notice that I said *can*, not 'do'. However, it behoves us as scientists to avoid importing wild claims into scientific articles, especially ones that may be published in journals such as GC. Despite our enthusiasm for simulation/gaming, we as scientists need to keep on board, in our intellectual pursuits, our healthy scepticism and distrust of mere claims. After all, that is why you are writing your article, to provide evidence that these things can work.

Some of your early text (abstract and intro) sounds like a commercial claim. In simulation/gaming, especially the commercial branches, it is easy to quote claims in someone's work, but if they are just claims, with little or no justification, then you are just repeating empty claims.

> Examples: "players are incentivised to understand climate-related concepts, adopt sustainable behaviours, and champion environmental stewardship", "immerses players in the multifaceted challenges of future climate scenarios, thereby deepening their appreciation of climate complexity".

We suggest that you check your sources to ascertain if they have provided credible, empirical support for their claims, or if they were inspired by their beliefs, hopes and enthusiasm – or simply repeated prior claims. When we run games, we can see whether or not participants are |

benefitting, especially if we run properly structured debriefing. However, in scientific publication, that would be considered as hearsay. We can also ask participants what they think that they learned, but self-report surveys (such as yours) are not often as strong as we would like them to be.

Quite a lot has been written about gamification, but much of it fails to include empirical support for its advantages. It would be good, in your lit rev, to distinguish publications that provide, from those that do not provide, empirical support for their gamification claims. It is fine to quote gamification scholars, and they provide clarity, but we still need to be wary of claims that are not backed up by some kind of reliable evidence.

One article that appears to provide some solid support is Sailer, M., Hense, J. U., Mayr, S. K., & Mandl, H. (2017). How gamification motivates: An experimental study of the effects of specific game design elements on psychological need satisfaction. *Computers in Human Behavior*, *69*, 371–380. The question is: How far can their conclusions be transferred to climate education?

This is one of the great difficulties in educational evaluation. It sometimes tries to accomplish the impossible task of emulating the more 'exact' sciences, like climatology or geology, which are able to use precise and objective measuring instruments. We are able to measure sea-level rise directly and extremely precisely, but we do not know how to measure knowledge rise directly or precisely, either in a game or in a traditional class. That is why efforts like yours are laudable – because they push forward both our instruments and the findings, in what is, frankly speaking, a rather murky world, compared with much of geoscience.

In regard to gamification, my hunch (sorry, it is not much more than that) is that little empirical work has been to provide accurate and reliable support for its effects. Much seems to be theorizing, supposition and argumentation, with a good dose of enthusiasm. We suggest that you examine the gamification lit carefully to see what refs provide empirical support and what refs provide theoretical, moral support. Some excellent early work was done by Landers (see the refs below). See also this lit rev Miao, H., Mohamad Saleh, M. S., & Zolkepli, I. A. (2022). Gamification as a Learning Tool for Pro-Environmental Behavior: A Systematic Review. *Malaysian Journal of Social Sciences and Humanities (MJSSH)*, *7*(12), e001881. Although Deterding et al has been widely cited, it is worth being careful, because

- o an early work, even if well cited, will have probably evolved,
- o it is a conference proceedings, and not a peer-reviewed journal,
- o the reader might begin to wonder if you are trying to promote their work, especially given that so much has been published since – the lit revs in the bibliography,
- o his affirmation that a game cannot contain gamification elements is simply not true.

| | |
|---|---|
| 35 knowledge action | The question of knowledge into action is a thorny question. In contrast to your two quotes, some research seems to indicate that knowledge does not lead to action – you can probably find refs to that work on the web. For a discussion in the context of simulation/gaming, see https://doi.org/10.1177/1046878108330364. |
| 98. Correct refs | In similar vein, we would suggest that you check all your refs for the content that you report. For example: You write "Gamification encourages learning and behaviour change through mechanisms like rewards, competition, and progression systems (DeWaters & Powers 2011, Kirchner et al. 2015)." Unfortunately, the DeWaters & Powers article does not mention |

gamification, gamification techniques such as rewards or even games. That is a rather serious error, and you should find other refs to support your statements, in this case, that gamification encourages learning etc. If possible those refs should contain empirical evidence, not just report opinion. If it is opinion, in articles or your own, then you need to say that it is opinion. Expressing opinion is not wrong, but it has to be stated, and if possible it should be informed or authoritative opinion, but it is nevertheless opinion.

This lit rev article on gamification should provide you with some good sources from which to draw, but still remain prudent: Miao, H., Mohamad Saleh, M. S., & Zolkepli, I. A. (2022). Gamification as a Learning Tool for Pro-Environmental Behavior: A Systematic Review. *Malaysian Journal of Social Sciences and Humanities (MJSSH)*, *7*(12), e001881.

| | |
|---|---|
| Games do not simulate, simulations do

Games include sims, and games are different from sims | "Games simulating real-world scenarios enable players to test strategies …", l.96. This sentence and the whole paragraph demonstrate the muddle in terminology that is manifest in much of the literature, and thus in our misunderstanding of the simulation/gaming field. In everyday situations, we tend to use the word 'game' when we are talking about things that are really simulations. This has been the habit and tradition in the simulation/gaming community since the 1980s. In fact, most so-called games about the climate are really simulations, usually those that include some form of representation of climate phenomena. In the more technical (and precise) uses, a simulation includes some form of representation of CC things; it may also include game elements, and even gamification techniques. In a scientific journal, it is good to stick with the more technical uses. That is why the WCS (EnRoads family) is correctly labelled.

In everyday discussion, the word game tends to cover a myriad of methods, techniques, activities (including simulations and role-plays). In what we like to consider as a more precise usage (in scientific articles, for instance), game and simulation are two very different beasts. You will find a tentative discussion in the reconceptualization article in the refs. |
| Debriefing | Given that your FG is in fact a game, or certainly an experiential leaning experience (see Kolb's experimental learning writings), albeit with gamification elements, it really does need to be fully debriefed – for a rational, see https://www.researchgate.net/publication/374344073_Debriefing_A_Practical_Guide.

If you did not do a formal de debriefing, then maybe some of the activities after the FG experience might be considered as (informal) debriefing. Even your post questionnaire could be considered as also providing a debriefing function. |
| Experts | Please say more about these experts and the guidance that they provided. After all, from what you say, they were crucial to the learning process. |
| Conclusion | We have not commented on this, as you will probably need to change it greatly in your next version. |
| CC | We suggest that you bring in much more about climate change, such as the aspects of CC that are pertinent to your FG, for example, the causes and effects, the physics, the social dimensions, etc. If you do not mention CC in a minimal way, then it would be difficult to justify publication in a geoscience journal, and publication in an educational journal would then be more appropriate. You might, for example, what specific CC things the students think that they learnt, what CC things were objectives of FG, what CC things you plan to include later. |

**Comments on research design**

| | |
|---|---|
| 126 features | "Finding Gaia incorporates several hallmark features of gamification to enhance engagement and learning (Deterding et al. 2011, Hamari et al. 2014):" I am not sure why you use the word 'hallmark' – In our view, it would sound better without. |
| 126 FG description | We find it difficult to get a sense of what FG is really like. You list five techniques used in GF. However, they sound like it is a list quoted from somewhere, rather than a graphic description of what actually **happens** in FG. For example, you say "Participants earn points or rewards for completing tasks, solving puzzles, or uncovering virtual treasures." It would be really good if you could say a little more about each of these. For example, what kinds of rewards (and how they help their learning), what types of puzzles (provide a couple of examples) and what types of treasures (provide examples). In addition, some pictures of these things would be very useful to the reader, and help them to understand your whole FG game. |
| | SWe have similar comments for the other four features. It would be useful too, to have some extracts of the live chats (or transcripts thereof). |
| | You say "leaderboards motivate participants", "points or rewards … foster[] competition", "riddles, puzzles, … add[] excitement and intrigue". Assuming that these apply to your FG game, do you have evidence of these things, eg, that leaderboards motivate, especially all participants? What about those who are less competitive, and you we do not notice because they are quiet and say little? |
| | At some point, maybe in section 2.2.1, we would very much welcome a full description of the game, including game objectives (we assume 'find the treasure'), learning goals (eg, understand how GHGs trap energy, understand why weather events bare more extreme than before, understand how CC affects poorer communities more than well-off ones, etc). Also indicate such things as: N° participants (players); age range, time for play, type of debriefing, time for debriefing, etc. |
| | It would be wonderful if you could provide a link to the game, so that readers can go and look for themselves, and thus get a far better idea of what your TH feels like. |
| 169-175 | This para does not belong here – it should go somewhere in your description/origin of the game. Spell out abbreviations, such as IDEE, on first use. |
| Gender & age | You say that you had roughly equal numbers of female and male students. As you have the data, this would be an excellent opportunity to analyse what differences exist between the two. For example, do women rate their knowledge, pre and post, significantly higher or lower than men? You would probably need to use the Mann-Whitney U test. This would greatly enrich your article. You could do the same for the three large age groups (maybe assimilating the small young group into the 16 year olds); if you have the actual ages in years and months, then you could split the whole group into two; if not, then just use the 15+16 and the 18 year-old groups. Assuming that age makes a difference, leaving out the 17 year olds would probably give you stronger results. |
| | If you have other demographic date, you could consider also using those. |

| Comments on method and stats |
| --- |

| 75 Method | In this section, you present the theory and advantages of gamification. In a research article, the methods section contains information about how you conducted the research (such as your measuring instruments), not about the object of research (your FG TH game). You should reorganize your material to separate the two very clearly, maybe into two broad sections, titled something like 'Gamification and FG' and then 'Evaluation'. Each of those can then have sub-sections.

All your text about gamification (its advantages and disadvantages, etc), about how you developed your FG TH game, about the components of the game, about the way you ran the game, etc, should be in early sections (eg, 'gamification' and 'development of FG' and 'use of the FG game').

All your text about how you evaluated, your evaluation instrument, the stats that you used, the results obtained, etc. should be in a whole new section (eg, Evaluation). This would include the standard sections for research. Here is a generic list, which you will have to adapt to your research:

  o  Identify and define an overall research objective, maybe with sub-objectives. In your case, this might be "do gamification techniques, embedded in a treasure hunt game, contribute to participants' understanding of CC concepts.

  o  Do a literature review: In your case, just a little on the advantages and drawbacks of gamification in general, and then a more in-depth look at gamification in CC literacy/education. If possible, identify research gaps, but your overall research project does not need this as you are evaluating a specific game. You can draw heavily on the two lit revs that have already been published on gamification.

  o  Explain your research design & methodology -- this includes your overall approach, including methods (qualitative, quantitative, mixed), participant selection, data collection instruments and procedures.

  o  Get ethical approval (if applicable).

  o  Collect data: Gather information according to the research design using the chosen instruments (e.g., surveys, tests, interviews, observations). You should provide a full copy of your research instrument ion the appendix. Explain how you used your questionnaire to get replies – online or on paper. Say the number that you sent out, and the number you got back.

  o  Analyse your data: Use appropriate statistical or qualitative techniques to identify patterns and themes. Specify what stats package you use, eg, sas, spss, r. In your case, we think that you should use a different test – see below.

  o  Interpret findings & draw conclusions: Make sense of the analysed data, relate findings back to the research question and literature, and acknowledge limitations. |
| 189 | You mention "control questions" before and after "the protocol". What are these control questions and what is THE protocol? |
| Stats | You say "t-Student test" – we think it should be "Student's t-test", but you can simply write t- |

test.  We think that it would be more appropriate to use a non-parametric equivalent of the t-test.  Looking at your graphs, your data do not appear to be parametric, which is usually the case in educational research.  A test, such as Wilcoxon Signed-Rank Test would be more appropriate – it is a nonparametric test for paired data.  It will probably give similar results, but your research could be criticised for using the wrong test.  The best thing would be to check with a good education statistics research scholar.  Although the use of t-tests on Likert scale data seems to be a common practice, it is statistically debatable due to the ordinal nature of the data and potential violations of the t-test's parametric assumptions.  You can look all that up on several stats websites.

We suggest that a nonparametric test might be better because they:

- o  make no assumption of normality:   Nonparametric tests do not assume that the data follows a normal distribution, which is often violated by Likert scale data.  Some of your bar charts seem to indicate a non-normal distribution.

- o  are based on ranks or frequencies:  They analyse the ranks of the data or the frequencies within categories, which is more aligned with the ordinal nature of a Likert scale the scale.

- o  do not mind outliers: Nonparametric tests are generally less sensitive to outliers compared to parametric tests.

We do not think that you need to worry about your results using Wilcoxon; generally you get similar results.  In addition, you avoid possible criticism later.

| 185 | The data (age percentages, gender distribution, prior experience percentages) are **not** qualitative; they are **quantitative**.  Qualitative data would involve descriptions, categories without numerical values, or themes. |
|---|---|

| 187 | The phrase "in fact" suggests that prior experience with treasure hunts is a subset of prior experience with escape rooms.  Although some overlap might exist, a treasure hunt is a distinct activity and cannot necessarily be considered an escape room.  This could be misleading. |
|---|---|

Rephrase to clarify the distinction.  For example: The majority of students reported no prior experience with recreational or educational escape rooms, and very few (only 9%) had participated in a treasure hunt before participating in FG.

| 191 | You write "To substantiate this observation and determine whether students assimilated the concepts presented, statistical correlation analyses of pre- and post-experience responses were conducted using a t-Student test to assess the significance of distribution differences." |
|---|---|

The stated goal is to determine if students "assimilated the concepts presented."  A paired test (t-test or Wilcoxon Signed-Rank test on Likert scale responses before and after participation in FG can show a statistically significant *change in perception* or *self-reported understanding*, but it does not directly measure *assimilation of concepts* in an objective sense.  Assimilation would typically require more direct assessment of knowledge retention or application.  This is one of the drawbacks of self-report studies.  One advantage is that you do not have to design more complex instruments that attempt assimilation more objectively.

Also, you mention "statistical correlation analyses", but your sentence then focuses solely on the t-test for distribution differences.  This creates a disconnect.  If you conducted correlation analyses and they are relevant to concept assimilation, you should discuss them more explicitly.

| 195 | You sentence "For each survey … of freedom" is a little problematic. The phrase "with a 99% probability ($p < 0.01$)" and "with a 95% probability ($p < 0.05$)" is slightly imprecise. The p-value represents the probability of observing the data if the null hypothesis were true. It is not the probability of the null hypothesis being false or the probability of the results being correct. |
|---|---|

To clarify this, we suggest that you rephrase to be more statistically accurate, something like this: For Questions A to D, the null hypothesis of no significant difference between pre- and post-protocol responses was rejected at a significance level of $\alpha = 0.01$ ($p < 0.01$). For Question E, the null hypothesis was rejected at a significance level of $\alpha = 0.05$ ($p < 0.05$). In both cases, the degrees of freedom were 206.

However, you may get slightly different numbers from the Wilcoxon Signed-Rank test.

| 201 | Although an increase in self-reported knowledge *suggests* the activity was perceived as educational and effective, it does not directly *confirm* it. Other factors influencing this perception might be at play. |
|---|---|

| 218 | Thus brings us up against the discussion (in your ms and in this review) of the 'nature' of gamification, of treasure hunt and of game. Contrary to what you seem to assume in your paper, we suggest that your FG treasure hunt as a whole is a game and that gamification techniques are used during the hunt game (presumably to make the game hunt ore appealing or engaging). |
|---|---|

Our perspective, and indeed the very nature of your research design, raises an anomaly in your sentences: "...indicating a more favourable perception. This suggests that integrating gamification elements into the virtual treasure hunt was regarded as a beneficial and appealing feature by the participants."

Here you discuss Question E, which asks about the effect of *virtual characteristic* on the treasure hunt experience. The conclusion that you draw is about the integration of *gamification elements*. Although the *virtual* format might have facilitated participation in your FG treasure hunt game, question E itself does not directly address *gamification*. This is a potential misinterpretation or a leap in inference.

You should ensure that the interpretation directly relates to the question asked. For example: 'This suggests that the virtual format of the treasure hunt was perceived more positively after the experience, indicating that the virtual elements did not detract from their engagement.' If gamification techniques were a key aspect, this connection should be made more explicit or supported by other data (potentially from qualitative data).

For all questions, we encourage you to ensure that the interpretations of each question's results directly address the content of the question asked. Be cautious about inferring specific mechanisms (like the impact of gamification elements) without direct evidence related to that aspect.

| 222 | This sentence refers to "escape room", while the previous sections primarily discussed a "virtual treasure hunt". This is a contradiction or at least a significant inconsistency in terminology. |
|---|---|

It is important to maintain consistent terminology throughout the article. If the experience involved elements of both, this needs to be clearly stated and differentiated. Choose one term

| | (e.g., "virtual educational treasure hunt with escape room elements") and use it consistently, or explain the relationship between the two. |
|---|---|
| 114 & 247 | You say that FG was about "climate change, including mitigation and adaptation strategies", but later you say that is for "imparting knowledge on seismic risks and climate change" and for the environment and geophysics; is it for all of those? That seems to be a bit of a tall order in a TH game. |
| 226 | Your sentences "They reported …" and "Moreover, the activity …" make claims that seem to us to be too strong. |
| | The first relies on self-reporting. Although this is valuable (and can be considered as a reasonable guide to suggest a positive outcome), it is not objective evidence of knowledge acquisition or application. It is always good to acknowledge that this is based on self-reported perceptions. You could, for example, say "According to their self-reports, students believed they acquired ...". |
| | The second is again based on student perception. Although perceptions are important, they are not a direct measure of team-building skill development or actual changes in academic interest. We suggest that you qualify this statement as based on student perceptions, for example, "Furthermore, students perceived the activity as beneficial for...". |
| Limitations | It is customary to include a section or what you consider to be the various limitations of your work, eg, maybe lack of debriefing, lack of linking specific gamification techniques to specific CC things. |
| | Your localized design (two events) may limit the applicability of the findings to other educational contexts or regions, as the unique characteristics of the participating institutions and their student populations may not reflect those of other schools or communities. This is why replications studies and future research is important. |
| Future research | It is also customary to suggest lines of research that you see following on from yours. |

**Inconsistencies, unclear antecedents or referents**

| | |
|---|---|
| Singular plural | Our primary **goal** was to position students as active participants, fostering scientific understanding alongside leadership and problem-solving skills. To achieve these **aims**, we employed |
| 59 | "Climate change education presents a compelling avenue for gamification, given its interdisciplinary nature and critical importance … **This method** simplifies c…" |
| | What does "this method" refer to? If it is gamification, I am not sure that I would call it a method; I tend to think of it as a collection of techniques or tools. |
| | What is it that is interdisciplinary? CC, education, gamification or all three? For a discussion in interdisciplinary in gaming, see https://doi.org/10.1177/104687810003100101 |
| 60 | This method simplifies complex concepts, making them accessible **through interactive games** that foster an understanding of climate systems |
| | 1. Are some games not interactive? – is that adjective necessary? |

2. If by 'this method', you mean gamification, then I a find it hard to understand how gamification can work through games; they can be a part of a game or simulation. I do not see how a gamification technique, such as a leader board, can simplify complex concepts. What is it that fosters? Is it games, which your grammar indicates, or is it gamification? If the later (or both), then you should rephrase.

You will find this useful: https://owl.purdue.edu/ -- it is considered one of the best guides to writing. For defining and non-defining clauses, see https://owl.purdue.edu/owl/general_writing/grammar/that_vs_which.html. See also the APA guides on the web.

| 164 & 170 replication | Your ms argues for the uniqueness and non-replicability of the your FG game as a strength, despite the following contradictions;
1. first, this seems to contradict earlier claims about establishing best practices;
2. second, it contradicts the fact that you replicated it a few months or a year later;
3. third, you mention scalability, which contradicts on-replicability;
4. fourth, a hallmark of good science is that it can be replicated, and that the research reports in sufficient detail for other people to be able to repeat your research. |
|---|---|

**Comments on presentation (tables & figures)**

| Table 1 | The more common form is Cronbach**'s** Alpha. |
|---|---|
| Table 2 | It would be good to include the actual questions for A to E – you can make the 2nd to 5th columns much narrower. Alternatively, put the questions just under the table.

It is far more common to use the terms 'mean', 'SD' and for p values a form like $p < .001$ (all your p values seem very small, but a non-parametric test may give larger values, but probably still $< .001$). |
| Table 3 | Similar comments |
| Fig 1 | See our comment about the 9% for escape rooms above. |
| Fig 2 | The title "How much and how do you …" is somewhat ambiguous. Was this the wording in the actual questions? Also, to help us understand better, you should clearly label the axes. We assume that the x axis is your Likert scale, but we have now idea of the labels used for each interval. It was at first unclear what the y axis represented – it needs to be labelled clearly – until we remembered that you had 206 students. It would be better to calculate in terms of percentage. You say "The figure compares", but maybe it should be something like 'Pre and post are compared in fig'. |
| Fig 3 | Label axes clearly. This type of fig not usually called a histogram. Probably better to call it a bar chart or a bar graph (each count in your Likert interval should be considered as a discrete number (category) – each response option (e.g., Strongly Disagree, Disagree, Neutral, etc) becomes a distinct category on the X-axis. |
| | It would probably be better if you could make your figs smaller, especially horizontally. They do not need to take up so much space. |
| Fig 7 | What are the actual questions for each of your three categories? |

| Fig 8 | Again, it would be good to spell out the actual question as it was given to the students, along with scale used. |

**Comments on English & terminology**

| not only but also | Avoid the structure "not only, but also", *not only A, but also B* reads more easily as *both A and B*. In any case, you MUST almost ALWAYS put a comma before but. |
| protocol | At first, we did not understand what you meant by protocol. The term protocol is used inconsistently throughout the paper, sometimes referring to the educational activity and sometimes to the research methodology. Better to use a simple term like 'our FG treasure hunt game' or 'our FL game', and then 'research protocol', which sounds better with 'research method'. |
| Other terms | Establish clear definitions for key terms at the outset and use them consistently throughout. |
| | Choose either "treasure hunt" or "escape room" and use it consistently, or clearly explain how they differ. |
| | Provide proper citation and explanation when introducing specialized concepts like "Science Capital". |
| alpha | The more common form is Cronbach**'s** Alpha. |

+++++++++++++

**Bibliography**

Albertazzi, D., Ferreira, M. G. G., & Forcellini, F. A. (2019). A Wide View on Gamification. *Technology, Knowledge and Learning*, *24*(2), 191–202.

Ansar, M., & George, G. (2023). Gamification in Education and Its Impact on Student Motivation—A Critical Review. In M. A. Chaurasia & C.-F. Juang (Eds.), *Emerging IT/ICT and AI Technologies Affecting Society* (pp. 161–170). Springer Nature.

Antonaci, A., Klemke, R., Stracke, C. M., & Specht, M. (2017). Gamification in MOOCs to enhance users' goal achievement. *2017 IEEE Global Engineering Education Conference (EDUCON)*, 1654–1662.

Bell, K. (2018). *Game on! Gamification, gameful design, and the rise of the gamer educator*. Johns Hopkins University Press.

Burke, B. (2014). *Gamify: How gamification motivates people to do extraordinary things*. Bibliomotion.

*Can't play, won't play*. (n.d.). hideandseek.net /2010/10/06/cant-play-wont-play

Chou, Y. (n.d.). *Actionable Gamification*. https://yukaichou.us5.list-manage.com/track/click?u=0e4d88e02b5b077b7bee2ca33&id=503b355f3b&e=db9834edac&i=3ea9aa3cb4

Christians, G. (2018). *The Origins and Future of Gamification* [University of South Carolina]. https://scholarcommons.sc.edu/senior_theses/254

Cook, J., Ecker, U. K. H., Trecek-King, M., Schade, G., Jeffers-Tracy, K., Fessmann, J., Kim, S. C., Kinkead, D., Orr, M., Vraga, E., Roberts, K., & McDowell, J. (2023). The cranky uncle game—Combining humor and gamification to build student resilience against climate misinformation. *Environmental Education Research*, *29*(4), 607–623.

Crookall, D., & et al. (1987). Towards a reconceptualization of simulation: From representation to reality. *Simulationj/Games for Learning*, *17*(4), 147–171. Get from https://www.researchgate.net/publication/284024653_Towards_a_Reconceptualization_of_Simulation_From_Representation_to_Reality

Deterding, S., Dixon, D., Khaled, R., & Nacke, L. (2011). From game design elements to gamefulness: Defining 'gamification'. *Proceedings of the 15th International Academic MindTrek Conference: Envisioning Future Media Environments*, 9–15.

Douglas, B. D., & Brauer, M. (n.d.). *Gamification to prevent climate change: A review of games and apps for sustainability*.

Dykens, I. T., Wetzel, A., Dorton, S. L., & Batchelor, E. (2021). Towards a Unified Model of Gamification and

Motivation. In R. A. Sottilare & J. Schwarz (Eds.), *Adaptive Instructional Systems. Design and Evaluation* (pp. 53–70). Springer International Publishing.

Fernández Galeote, D., Rajanen, M., Rajanen, D., Legaki, N.-Z., Langley, D. J., & Hamari, J. (2021). Gamification for climate change engagement: Review of corpus and future agenda. *Environmental Research Letters*, *16*(6), 063004.

Ferrara, J. (2013). Games for Persuasion: Argumentation, Procedurality, and the Lie of Gamification. *Games and Culture*, *8*(4), 289–304.

Gamification. (2025). In *Wikipedia*. https://en.wikipedia.org/w/index.php?title=Gamification&oldid=1288873006

*Gamification: The Application of Game Design in Everyday Life*. (n.d.).

Goethe, O., & Palmquist, A. (2020a). Broader Understanding of Gamification by Addressing Ethics and Diversity. In C. Stephanidis, D. Harris, W.-C. Li, D. D. Schmorrow, C. M. Fidopiastis, P. Zaphiris, A. Ioannou, X. Fang, R. A. Sottilare, & J. Schwarz (Eds.), *HCI International 2020 – Late Breaking Papers: Cognition, Learning and Games* (pp. 688–699). Springer International Publishing.

Goethe, O., & Palmquist, A. (2020b). Broader Understanding of Gamification by Addressing Ethics and Diversity. In C. Stephanidis, D. Harris, W.-C. Li, D. D. Schmorrow, C. M. Fidopiastis, P. Zaphiris, A. Ioannou, X. Fang, R. A. Sottilare, & J. Schwarz (Eds.), *HCI International 2020 – Late Breaking Papers: Cognition, Learning and Games* (Vol. 12425, pp. 688–699). Springer International Publishing.

Hou, H. T. (Ed.). (2023). *Game-Based Learning and Gamification for Education*. MDPI - Multidisciplinary Digital Publishing Institute.

Johnson, D., Deterding, S., Kuhn, K.-A., Staneva, A., Stoyanov, S., & Hides, L. (2016). Gamification for health and wellbeing: A systematic review of the literature. *Internet Interventions*, *6*, 89–106.

Kapp, K. (n.d.). *Games, Gamification and the Freedom to Fail*.

Kapp, K. M. (n.d.). *The Gamification of Learning and Instruction: Game-Based Methods and Strategies for Training and Education*.

Kim, S., Song, K., Lockee, B., & Burton, J. (2018). *Gamification in Learning and Education*. Springer International Publishing.

Kim, T. W., & Werbach, K. (2016). More than just a game: Ethical issues in gamification. *Ethics and Information Technology*, *18*(2), 157–173.

Krath, J., Schürmann, L., & Von Korflesch, H. F. O. (2021). Revealing the theoretical basis of gamification: A systematic review and analysis of theory in research on gamification, serious games and game-based learning. *Computers in Human Behavior*, *125*, 106963.

Kriz, W. C., Kikkawa, T., & Sugiura, J. (2022). Manipulation Through Gamification and Gaming. In T. Kikkawa, W. C. Kriz, & J. Sugiura (Eds.), *Gaming as a Cultural Commons: Risks, Challenges, and Opportunities* (pp. 185–199). Springer Nature.

Landers, R. N. (n.d.). *Developing a Theory of Gamified Learning: Linking Serious Games and Gamification of Learning*.

Landers, R. N., Armstrong, M. B., & Collmus, A. B. (2017). How to Use Game Elements to Enhance Learning: Applications of the Theory of Gamified Learning. In M. Ma & A. Oikonomou (Eds.), *Serious Games and Edutainment Applications: Volume II* (pp. 457–483). Springer International Publishing.

Landers, R. N., Bauer, K. N., Callan, R. C., & Armstrong, M. B. (2015). Psychological Theory and the Gamification of Learning. In T. Reiners & L. C. Wood (Eds.), *Gamification in Education and Business* (pp. 165–186). Springer International Publishing.

Landers, R. N., & Callan, R. C. (2011). Casual Social Games as Serious Games: The Psychology of Gamification in Undergraduate Education and Employee Training. In M. Ma, A. Oikonomou, & L. C. Jain (Eds.), *Serious Games and Edutainment Applications* (pp. 399–423). Springer London.

Lee, J. J., & Hammer, J. (n.d.). *Gamification in Education: What, How, Why Bother?*

Liu, Y. (2024a). Gamification in Climate Action: Understanding the Role of Game Technologies and Participatory Engagement. In I. Hossain, A. K. M. M. Haque, & S. M. A. Ullah (Eds.), *Advances in Environmental Engineering and Green Technologies* (pp. 359–380). IGI Global.

Liu, Y. (2024b). Gamification in Climate Action: Understanding the Role of Game Technologies and Participatory Engagement. In I. Hossain, A. K. M. M. Haque, & S. M. A. Ullah (Eds.), *Advances in Environmental Engineering and Green Technologies* (pp. 359–380). IGI Global.

Luo, Z. (2022). Gamification for educational purposes: What are the factors contributing to varied effectiveness? *Education and Information Technologies*, *27*(1), 891–915.

Miao, H., Mohamad Saleh, M. S., & Zolkepli, I. A. (2022). Gamification as a Learning Tool for Pro-Environmental Behavior: A Systematic Review. *Malaysian Journal of Social Sciences and Humanities (MJSSH)*, *7*(12), e001881.

Niman, N. B. (2014). *The Gamification of Higher Education*. Palgrave Macmillan US.

Ouariachi, T., Li, C.-Y., & Elving, W. J. L. (2020). Gamification Approaches for Education and Engagement on Pro-Environmental Behaviors: Searching for Best Practices. *Sustainability*, *12*(11), 4565.

Reiners, T., & Wood, L. C. (Eds.). (2015). *Gamification in Education and Business*. Springer International Publishing.

Sailer, M., Hense, J. U., Mayr, S. K., & Mandl, H. (2017a). How gamification motivates: An experimental study of the

effects of specific game design elements on psychological need satisfaction. *Computers in Human Behavior*, *69*, 371–380.

Sailer, M., Hense, J. U., Mayr, S. K., & Mandl, H. (2017b). How gamification motivates: An experimental study of the effects of specific game design elements on psychological need satisfaction. *Computers in Human Behavior*, *69*, 371–380.

Satyagraha, A., Lassa, J., Amri, A., Yoliando, F. T., Herna, L., & Killing, I. Y. (2025). *Gaming and Gamification Approach as Pathways to Sustain Climate Change and Disaster Education in Low- and Middle-Income Countries: Reflection in Action and on Action from Generaksi*. SSRN.

Simões, J., Redondo, R. D., & Vilas, A. F. (2013). A social gamification framework for a K-6 learning platform. *Computers in Human Behavior*, *29*(2), 345–353.

Stieglitz, S., Lattemann, C., Robra-Bissantz, S., Zarnekow, R., & Brockmann, T. (Eds.). (2017). *Gamification*. Springer International Publishing.

Thibault, M., & Hamari, J. (2021). Seven Points to Reappropriate Gamification. In A. Spanellis & J. T. Harviainen (Eds.), *Transforming Society and Organizations through Gamification: From the Sustainable Development Goals to Inclusive Workplaces* (pp. 11–28). Springer International Publishing.

Tondello, G. F., Mora, A., Marczewski, A., & Nacke, L. E. (2019). Empirical validation of the Gamification User Types Hexad scale in English and Spanish. *International Journal of Human-Computer Studies*, *127*, 95–111.

van Roy, R., & Zaman, B. (2017). Why Gamification Fails in Education and How to Make It Successful: Introducing Nine Gamification Heuristics Based on Self-Determination Theory. In M. Ma & A. Oikonomou (Eds.), *Serious Games and Edutainment Applications: Volume II* (pp. 485–509). Springer International Publishing.

What "Gamification" is and what it's not. (2017a). *European Journal of Contemporary Education*, *6*(2).

What "Gamification" is and what it's not. (2017b). *European Journal of Contemporary Education*, *6*(2).

Wolf, T. (n.d.). *Green gamification: How gamified information presentation affects pro-environmental behavior*.

Woodcock, J., & Johnson, M. R. (2018). Gamification: What it is, and how to fight it. *The Sociological Review*, *66*(3), 542–558.

Zhang, X. (2023). Climate Change Literacy Gamified: How Gamification Mechanics Affect User Experience Factors in the User Interfaces. In J. Y. C. Chen, G. Fragomeni, & X. Fang (Eds.), *HCI International 2023 – Late Breaking Papers* (Vol. 14058, pp. 408–425). Springer Nature Switzerland.

---

## Editor Comment (EC1)

**Referee report for the manuscript egusphere-2025-577**

**Finding Gaia: Exploring Climate Change Through Gamification**

**General comments**

The manuscript presents the results of an educational experience focused on climate change, which follows a gamification approach. The paper provides information on the context and the approach used in this study and the main findings and it discusses issues around using this approach to communicate science and raise awareness. It introduces a gamified educational experience and evaluates its impact using statistical methods.

The topic of the manuscript is valuable for scientists seeking ways to communicate complex concepts to non-scientists, particularly young individuals. The adopted approach reflects a growing interest in alternative methods for engagement within an educational setting, and the study's outcomes provide insights into how communication can shape people's understanding and, consequently, influence awareness.

The manuscript is well written and organised, with a logical flow from background to methodology, results, and discussion. The narrative is engaging and suitable for a broad audience.

However, the manuscript should be improved to bring out the valuable points it discusses, increasing the impact of the study and broadening the readership. The proposed enhancements will broaden the reach and relevance of the paper and better support its purpose.

As a recommendation, the paper should benefit from revisions before being accepted for publication.

Specific suggestions on how the manuscript can be improved are included below.

**Specific comments**

1. The manuscript does not clearly define the research aim and objectives; is it to create the game, to evaluate its impact or to highlight the contribution of gamification in geosciences communication? The authors should clearly point out the aim of the study early in the manuscript, ideally at the end of the introduction, to guide the readers, allowing them to to better understand the process and the value of the empirical findings.

2. While the authors provide some information on the features of the activity, they do not include details on the content and learning mechanisms. For example, while they state that the activity focuses on climate change, including mitigation and adaptation strategies, they do not offer examples to allow the readers to understand the type of questions the activity includes. Including examples of puzzles or challenges and how they were used to address specific climate topics (for example, flooding and adaptation strategies) would provide more insight into how these topics were communicated through gamification.

3. Adding to the previous point, the title of the manuscript suggests a focus on exploring climate change through an educational activity that incorporates game-like elements. Although the reader can make a connection when examining the outcomes, the activity's approach to addressing climate change is not clearly presented. Strengthening this connection between the thematic context and the activity design would enhance clarity and value for the reader.

4. The authors describe the experience as not replicable, stating this as a limitation and at the same time as a unique characteristic. I think this point needs further explanation. Why is it not replicable? What specific elements constrain reproducibility? The authors could elaborate on this to help the reader understand its uniqueness. Also, the authors could discuss how the approach can be adapted/amended for implementation in other educational settings, which would increase the study's applicability and impact.

5. Building on the above point, the paper could benefit from a more critical discussion of the limitations, such as generalisability.

6. While the abstract of the manuscript provides valuable details on the context of the study, it only presents limited information on the study itself. I recommend the authors revise this to communicate the study's scope, methods better and highlight the key findings. This will improve the visibility of the work.

7. The manuscript uses different terms (for example, "gamification") that may be unfamiliar to the typical readers. I recommend defining these terms early in the manuscript to eliminate any confusion.

8. In the discussion section, the authors note that when single-session activities are complemented by sustained engagement over a longer period, they can support retention of complex topics. This is a valuable point that deserves further elaboration. Providing more information to support this statement would enable the readers to understand the potential of this study.

This is a promising and well-written manuscript that presents a novel and meaningful contribution. I strongly believe the above points would maximise its impact.